# Nonequilibrium Thermodynamics of the Majority Vote Model

**DOI:** 10.3390/e25081230

**Published:** 2023-08-18

**Authors:** Felipe Hawthorne, Pedro E. Harunari, Mário J. de Oliveira, Carlos E. Fiore

**Affiliations:** 1Instituto de Física, Universidade de São Paulo, Rua do Matão, 1371, São Paulo 05508-090, SP, Brazil; felipehgc@gmail.com (F.H.); oliveira@if.usp.br (M.J.d.O.); 2Complex Systems and Statistical Mechanics, Department of Physics and Materials Science, University of Luxembourg, L-1511 Luxembourg, Luxembourg; pedroharunari@gmail.com

**Keywords:** majority vote models, stochastic thermodynamics, phase transitions and spin systems

## Abstract

The majority vote model is one of the simplest opinion systems yielding distinct phase transitions and has garnered significant interest in recent years. This model, as well as many other stochastic lattice models, are formulated in terms of stochastic rules with no connection to thermodynamics, precluding the achievement of quantities such as power and heat, as well as their behaviors at phase transition regimes. Here, we circumvent this limitation by introducing the idea of a distinct and well-defined thermal reservoir associated to each local configuration. Thermodynamic properties are derived for a generic majority vote model, irrespective of its neighborhood and lattice topology. The behavior of energy/heat fluxes at phase transitions, whether continuous or discontinuous, in regular and complex topologies, is investigated in detail. Unraveling the contribution of each local configuration explains the nature of the phase diagram and reveals how dissipation arises from the dynamics.

## 1. Introduction

Opinion dynamics is a crucial issue in sociophysics, encompassing several topics, such as complex social processes, populational dynamics, decision making, elections and spreading of fake news/rumors and others [1]. In recent years, distinct approaches have been proposed and investigated, aimed at tackling the key aspects of opinion dynamics. Several of them deal with systems presenting phase transitions, marking the existence of two regimes, one of which has a prevailing given opinion.

Among the distinct opinion systems, the majority vote (MV) model is highlighted by its simplicity and for exhibiting universal features of nonequilibrium phase transitions. Its interaction mechanism comprises the agent’s tendency to align (follow) its opinion based on the majority opinion of its nearest neighborhood [2,3,4]. Subsequently, generalizations of the MV model have aroused interest, taking into account the influence of network topology [3,4], the inclusion of distinct noises [5,6], more states per agents [7,8] and more recently, inertial effects [8,9,10]. In the latter, the presence of inertial terms has revealed a robust mechanism, affecting the classification of phase transitions even for different lattice topologies [10] and in systems subject to temporal disorder [11]. More recently, the main properties of the MV have been extensively studied in terms of entropy production signatures [12,13].

Stochastic thermodynamics [14,15,16,17] has become one of the most important topics in the realm of nonequilibrium statistical mechanics and an appropriate framework for describing the thermodynamic properties of Markovian nonequilibrium systems, having as a starting point a suitable definition of production of entropy that is able to discern equilibrium from nonequilibrium systems [18]. It presents intrinsically out of equilibrium results, such as generalizations of the first and second laws, that can be used to describe dynamics in terms of energy exchanges, their fluctuations and limits. Despite previous works investigating the main properties of the MV through the entropy production [12,13], other quantities have to be defined to establish a firm link between voter models and stochastic thermodynamics’ own framework. Aimed at overcoming such a drawback, a thermodynamic description for opinion models has been recently proposed [19], in which the idea of a distinct thermal reservoir per neighborhood opinion configuration was introduced. Such a framework not only allows one to associate the dynamics with well-defined temperatures but also reconciles the relationship between entropy production and heat flux.

In this paper, we advance on the aforementioned idea, by thoroughly investigating the thermodynamics of the majority vote model. More concretely, a general and unambiguous temperature definition is derived, providing a way to properly investigate the behavior of entropy production and heat fluxes in distinct phases as well as in continuous and discontinuous transition regimes. The investigation is also aimed at understanding the roles of inertia and distinct topologies (regular and complex), and testing fluctuation theorems. Given that the number of reservoirs and heat fluxes increases with the connectivity, the analysis of their roles and which of them is more representative of entropy production will be addressed and, finally, used to probe its traits across phase transitions.

These introduced thermal reservoirs and their respective temperatures are of an effective nature rather than physical entities. They are introduced to accommodate models that otherwise would not benefit from stochastic thermodynamics’ toolbox, thus extending the validity of its results.

This paper is organized as follows: Section 2 introduces the model and its main properties, model thermodynamics are presented in Section 3 and conclusions are drawn in Section 4.

## 2. Majority Vote Model and Phase Transition Behavior

In this section, we present an overview of the majority vote model and its phase transition aspects. It consists of a simple system with $Z_{2}$ “up-down” symmetry, in which each microscopic configuration η is set by the collection of *N* individuals $\eta \equiv (\eta_{1},\eta_{2},...,\eta_{i},...,\eta_{N})$, with $\eta_{i}$ being the spin variable of site *i* which takes the values ±1 according to whether the spin is “up” or “down”. With probability 1−f, the spin $\eta_{i}$ tends to align itself with its local neighborhood majority. Conversely, with complementary probability *f*, the majority rule is not followed. The inertial version differs from the original one by the inclusion of a term proportional to the local spin competing with the neighborhood. The model dynamics are governed by the following master equation
(1)$$\frac{d}{dt}P\left(\eta ,t\right)=\sum_{i=1}^{N}\left\{w_{i}\left(\eta^{i}\right)P\left(\eta^{i},t\right)-w_{i}\left(\eta \right)P\left(\eta ,t\right)\right\},$$
where $w_{i}\left(\eta \right)$ comprises the transition rate at which each site *i* changes its opinion from $\eta_{i}$ to $-\eta_{i}$, given by
(2)$$w_{i}\left(\eta \right)=\frac{1}{2}\left\{1-\left(1-2f\right)\eta_{i}\mathrm{sgn}\left(X\right)\right\},$$
where $X=\left(1-\theta \right)\ell +k\theta \eta_{i}$, *k* is the connectivity of a site, θ is the inertia strength, and sgn(X) is the sign function. The term *ℓ* plays a key role in the following results; it is defined as the sum of a site’s neighboring spins, *ℓ* := $\sum_{j}\eta_{j}$, and we omit the dependence on η for convenience. At a given configuration, all sites with *ℓ* will become thermodynamically equivalent, defining a thermal reservoir. The system presents two ferromagnetic phases for small *f*. Upon raising *f*, the system yields an order–disorder phase transition, where the value of the critical point is dependent on the lattice topology and neighborhood [2,3]. Although phase transitions are always continuous for the original model (θ=0) [4], the inclusion of inertia can shift the phase transition from continuous to discontinuous depending on the lattice topology and the neighborhood [8,9,10].

Since Equation (Equation 2) states that the transition rate depends on the sign of *X*, the flip probability (whether 1−f or *f*) will depend on the interplay between the number of nearest neighbors *k* and θ. For example, for $\eta_{i}=-1$, the argument of sgn(X) reads ℓ−θ(k+ℓ) implying that the transition −1→+1, due to a neighborhood with ℓ/(k+ℓ)>θ, occurs with probability 1−f (similar to the inertialess case), whereas when ℓ/(k+ℓ)<θ the probability is *f*. Thus, we define $\ell^{*}:=-k\theta \eta_{i}/\left(1-\theta \right)$ as the threshold value splitting the neighborhoods: All values $\left|\ell |>\right|\ell^{*}|$ yield a transition rate equals to the inertialess case, while $\left|\ell |<\right|\ell^{*}|$ yield $w_{i}\left(\eta \right)=f$ regardless of $\eta_{i}$. For completeness, the transition rate is 1/2 when both values are the same.

For fixed *k*, as in the present case, the phase diagram θ versus *f* will be characterized by plateaus. If θ is increased, the plateaus emerge when $\ell^{*}$ has an even integer value, since it marks a regime where one additional neighborhood type *ℓ* shifts its contribution f↔1−f. The plateaus can be obtained by relation
(3)$$\theta^{*}=\frac{2m}{k+2m},\;m\in N.$$
For instance, when the connectivity is k=20, these values are
(4)$$\theta^{*}=\left\{\frac{1}{11},\frac{1}{6},\frac{3}{13},\frac{2}{7},\frac{1}{3},\frac{3}{8},\frac{7}{17},\frac{4}{9},\frac{9}{19},\frac{1}{2}\right\},$$
which are later verified in the phase diagrams obtained by simulations in the first figure in Section 2.2 and the first figure in Section 2.3. They are the same for both regular and random-regular topologies, although the classification of the phase transition is also influenced by the topology, demonstrating that the mechanism behind the appearance of such plateaus is related to sharp shifts of contribution of each neighborhood *ℓ*.

### 2.1. Entropy Production

The entropy production and its connection with the heat flux is the central issue of this paper. Before relating both of them, we first review the main features of the microscopic entropy production formula. Starting with the entropy definition S=−〈lnP(η)〉 (here and hereafter, we adopt the convention $k_{B}=1$ for the Boltzmann constant) and assuming the system is in contact with a (or multiple) reservoir(s), its time derivative dS/dt is the difference between two terms: dS/dt=Π−σ, where Π and σ are the entropy production and entropy flux rates, given by the generic expressions:(5)$$\Pi =\frac{1}{2}\sum_{\eta }\sum_{i}\left\{w_{i}\left(\eta^{i}\right)P\left(\eta^{i},t\right)-w_{i}\left(\eta \right)P\left(\eta ,t\right)\right\}\ln\frac{w_{i}\left(\eta^{i}\right)P\left(\eta^{i},t\right)}{w_{i}\left(\eta \right)P\left(\eta ,t\right)}$$
and
(6)$$\sigma =\frac{1}{2}\sum_{\eta }\sum_{i}\left\{w_{i}\left(\eta^{i}\right)P\left(\eta^{i},t\right)-w_{i}\left(\eta \right)P\left(\eta ,t\right)\right\}\ln\frac{w_{i}\left(\eta^{i}\right)}{w_{i}\left(\eta \right)}.$$
where the one-site dynamics assumption was used. Since dS/dt=0 in the nonequilibrium steady state (NESS), in which $P\left(\eta ,t\right)\to p^{\mathrm{st}}\left(\eta \right)$, the steady entropy production can be calculated from σ, acquiring the convenient ensemble average form [13]:(7)$$\sigma =\sum_{i}\langle w_{i}\left(\eta \right)\ln\frac{w_{i}\left(\eta \right)}{w_{i}\left(\eta^{i}\right)}\rangle ,$$

In order to evaluate σ from Equation (Equation 7) we take the ratio between $w_{i}\left(\eta \right)$ and its reverse $w_{i}\left(\eta^{i}\right)$ given by
(8)$$\frac{w_{i}\left(\eta \right)}{w_{i}\left(\eta^{i}\right)}=\frac{1-\left(1-2f\right)\eta_{i}\mathrm{sgn}\left[\left(1-\theta \right)\ell +k\theta \eta_{i}\right]}{1+\left(1-2f\right)\eta_{i}\mathrm{sgn}\left[\left(1-\theta \right)\ell -k\theta \eta_{i}\right]}.$$

Inspection of the ratio above reveals that only local configurations where $\left|\ell |>\right|\ell^{*}|$ will contribute to the entropy production. When $\left|\ell |<\right|\ell^{*}|$, the ratio vanishes and therefore these configurations yield reversible transitions. This property is illustrated in Figure 1, where values of $\ell^{*}$ are shown in terms of θ. For a given finite *k*, the even values of *ℓ* locate the plateaus in the figure.

For ℓ≠kθ/(1−θ), Equation (Equation 8) is conveniently rewritten as
(9)$$\ln\frac{w_{i}\left(\eta \right)}{w_{i}\left(\eta^{i}\right)}=-\eta_{i}\mathrm{sgn}\left(\ell \right)H\left[\left|\ell \right|-\frac{k\theta }{1-\theta }\right]\ln\left(\frac{1-f}{f}\right),$$
where H(•) is the Heaviside function. However, for ℓ=kθ/(1−θ), marking the plateau position, Equation (Equation 9) acquires a distinct value given by $\ln\left(w_{i}\left(\eta \right)/w_{i}\left(\eta^{i}\right)\right)=\eta_{i}\mathrm{sgn}\left(\ell \right)\ln\left(2f\right)$.

The above formulae are equivalent by calculating such a ratio only over the subspace of local configurations in which the ratio is different from 1, that is for ℓ≥kθ/(1−θ) [13]. Thus, when expressed in terms of the misalignment parameter *f*, the steady entropy production σ is given by
(10)$$\begin{array}{cc} \sigma =\frac{1}{2}\ln\frac{1-f}{f}\{ & \left(1-2f\right)\langle {\mathrm{sgn}}^{2}\left(\ell \right)H\left[\left|\ell \right|-\frac{k\theta }{1-\theta }\right]\rangle \\ \; & -\langle \eta_{i}\mathrm{sgn}\left(\ell \right)H\left[\left|\ell \right|-\frac{k\theta }{1-\theta }\right]\rangle \}, \end{array}$$
which only depends on *f* and on $\langle \eta_{i}\mathrm{sgn}\left(\ell \right)H\left[|\ell |-k\theta /(1-\theta )\right]\rangle$ and $\langle {\mathrm{sgn}}^{2}\left(\ell \right)H\left[|\ell |-k\theta /(1-\theta )\right]\rangle$.

### 2.2. Overview about Phase Transitions and Finite-Size Scaling

As stated broadly in the literature, continuous and discontinuous phase transitions become rounded at the vicinity of phase transitions due to finite size effects, whether for equilibrium [20,21] and nonequilibrium systems [22,23]. Despite the order parameter and its moments have been broadly exploited for characterizing nonequilibrium phase transitions, recently, the behavior of entropy production and allied quantities (e.g., its first derivative) has attracted a great deal of attention as their identificators [12,13,24,25,26,27].

According to the finite-size scaling (FSS) theory, at the vicinity of the critical point $f_{c}$, a given quantity *X* [X∈(|m|,χ and $\sigma^{\prime }\equiv d\sigma /df)$] will behave as $X=N^{y_{x}/\nu }f_{x}(N^{1/\nu }\left|\epsilon |\right)$, where $f_{x}$ is a scaling function, $\epsilon =\left(f-f_{c}\right)/f_{c}$ is the distance to the criticality and $y_{x}$ is the critical exponent obtained from ($y_{x}=-\beta ,\gamma$ and α) [20]. The last exponent is similar to the relationship between the thermal derivative of the entropy, *S*, and specific heat, *C*, in equilibrium phase transitions (recalling that $C=N^{\alpha /\nu }f_{c}(N^{1/\nu }\left|\epsilon \right|$) [20], illustrating that the connection between entropy production and exchanged heat presented here introduces a physical argument for such scaling behavior.

Since the scaling behavior of heat fluxes (and their derivatives) at the criticality was considered previously in [19] we are going to focus on nonequilibrium discontinuous phase transitions in this paper. For a generic ensemble average *X*, the starting point consists of assuming a bimodal Gaussian distribution, centered at $\mu_{o}$ and $\mu_{d}$ (with associated variances $\chi_{o}$ and $\chi_{d}$). In the case of the steady entropy production at the vicinity of $\epsilon =f-f_{c}$, a bimodal entropy production probability distribution centered at $\mu_{o}$ and $\mu_{d}$ (with associate variances $\chi_{d}$ and $\chi_{o}$) leads to the approximate expression for σ:(11)$$\sigma \approx \frac{\mu_{o}+\bar{\alpha }\mu_{d}e^{-N[\left(\mu_{o}-\mu_{d}\right)\epsilon ]}}{1+\bar{\alpha }e^{-N[\left(\mu_{o}-\mu_{d}\right)\epsilon ]}},$$
where $\bar{\alpha }=\sqrt{\chi_{d}/\chi_{o}}$. We note that the ordered and disordered phases are favored as ϵ<0 and ϵ>0 (assuming that $\mu_{o}<\mu_{d}$), respectively, and $\sigma =\left(\mu_{o}+\bar{\alpha }\mu_{d}\right)/\left(1+\bar{\alpha }\right)$ at ϵ=0, indicating that all entropy production curves, simulated for distinct *N*’s, will cross at the transition point $f_{c}$. Having σ, its derivative in respect to *f* behaves at the vicinity of $f_{c}$ as:(12)$$\sigma^{\prime }\approx \frac{N{\left(\mu_{o}-\mu_{d}\right)}^{2}e^{N(\mu_{o}-\mu_{d})\epsilon }}{\bar{\alpha }{\left(1+\bar{\alpha }e^{N(\mu_{o}-\mu_{d})\epsilon }\right)}^{2}},$$
showing that $\sigma^{\prime }$ scales with *N* at the coexistence ϵ=0, in agreement with the above finite size expression for the quantity *X*. Alternatively (and analogously), Equation (Equation 11) is obtained by resorting to the ideas presented in [28,29,30], where coexisting phases are treated via a two-state model in which ordered and disordered phases are given by transition rates exhibiting an exponential dependence on the system size *N* and proportionality to the distance ϵ to the transition point:(13)$$a∼k\sqrt{\chi_{a}}e^{-N(c_{0}-c_{a}\epsilon )},\;b∼k\sqrt{\chi_{b}}e^{-N(c_{0}+c_{b}\epsilon )},$$
where $k,c_{0},c_{a},c_{b}>0$ are constants. “Ordered” and “disordered” probabilities, *p* and *q*, respectively, are related to rates *a* and *b* by means of relations p=b/(a+b) and q=1−p, given by $p=\sqrt{\chi_{b}}{\left(\sqrt{\chi_{b}}+\sqrt{\chi_{a}}e^{cN\epsilon }\right)}^{-1}$, where $c=c_{a}+c_{b}>0$. As shown in Ref. [29], a given ensemble average including the entropy production $\sigma =\langle \sigma_{\tau }\rangle /\tau$ averaged over a sufficiently long time t→∞ and over many independent stochastic trajectories given by $\sigma =\mu_{a}p+\mu_{b}b$, where
(14)$$\sigma \approx \frac{\mu_{b}\sqrt{\chi_{b}}+\mu_{a}\sqrt{\chi_{a}}e^{cN\epsilon }}{\sqrt{\chi_{b}}+\sqrt{\chi_{a}}e^{cN\epsilon }},$$
which has precisely the form of Equation (Equation 11).

The main features of discontinuous phase transitions are summarized in Figure 2. From now on we shall consider k=20 which exhibits a discontinuous phase transition for θ>1/3, as depicted in panel Figure 2a). Aforementioned portraits are exemplified in panels (b)−(d) for θ=3/8. We remark that continuous lines, given by Equation (Equation 11), describe very well the behavior of the entropy production and its derivative, the latter presenting a maximum at $f_{c}^{*}$ scaling with $N^{-1}$, whose value as N→∞ agrees very well with those obtained from the crossing among curves.

### 2.3. Discontinuous Phase Transitions in Complex Topologies

The behavior of discontinuous phase transitions in complex topologies is more revealing and it is different for small and large system sizes. In the former case, quantities change smoothly as *f* is varied (see e.g., Figure 3c), in similarity with the behavior in regular structures, also characterized by the reduced cumulant $U_{4}$ presenting a minimum value increasing with *N* (inset) and a maximum behavior of χ near the coexistence. Conversely, the behavior becomes akin to the mean-field when *N* is large, in which the phase coexistence manifests itself by means of a hysteretic branch, e.g., a region located at $f_{b}<f<f_{c}$ when the dynamics evolve to the ordered (stable for $f\leq f_{b}$) and disordered (stable for $f\geq f_{c}$) phases depending on the initial condition. Such changes upon raising *N* share some similarities with the metastable behavior observed in the dynamics and thermodynamics of work-to-work transducers, where the system behavior “quickly” approaches the MFT’s [behavior] as *N* increases [31].

Here, we describe a brief (nonrigorous) argument about the expected behavior in complex topologies by resorting to the ideas from Ref. [13]. Since spins are independent of each other in the disordered phase, the order parameter behaves as $\langle \eta_{i}\rangle ∼N^{-1/2}$ and then a *n*-th correlation will behave as $\langle \eta_{i}\eta_{i+1}...\eta_{i+n}\rangle \approx \langle \eta_{i}\rangle \langle \eta_{i+1}\rangle ...\langle \eta_{i+n}\rangle =N^{-n/2}$. Hence, in the thermodynamic limit, all correlations will vanish and σ will depend solely on control parameters. On the other hand, $\langle \eta_{i}\eta_{i+1}...\eta_{i+n}\rangle$ is expected to be finite and also f− dependent in the ordered phase, consistent with σ exhibiting a dependence on the control parameters and correlations. Therefore, the existence of a hysteretic loop for the order parameter (panel (b)) is also translated to the entropy production behavior (see e.g., panel (d)).

## 3. Thermodynamics of the Majority Vote Model

### 3.1. General Features

In Section 2, the main properties of the majority vote model were analyzed without any thermodynamic consideration. Here, we incorporate the notion of temperatures mediating changes of configuration in order to establish the connection with thermodynamics. Since the stochastic dynamics over the configuration space is fully determined by its transition rates, we shall resort the ideas from Refs. [16,17,32] in which transition rates are defined by assuming the local detailed balance. The central point consists of assuming that the one-site transition rate $w_{i}\left(\eta \right)$ is decomposed in *ℓ* distinct (and mutually exclusive) components, each one associated with a given thermal reservoir (reciprocal inverse temperature $\beta_{\ell }$), given by $w_{i}\left(\eta \right)=\sum_{\ell }w_{\ell i}\left(\eta \right)$ (ℓ=2,4,...,k), where $w_{\ell i}\left(\eta \right)$ assumes the Glauber form:(15)$$w_{\ell i}\left(\eta \right)=\frac{\alpha_{\ell }}{2}\left\{1-\tanh\left(\beta_{\ell }\Delta E\right)\right\},$$
where $\alpha_{\ell }$ is a constant and $\Delta E=E\left(\eta^{i}\right)-E\left(\eta \right)$ denotes the energy difference between configurations η and $\eta^{i}$. For “up-down” $Z_{2}$ symmetry systems, the energy can be generically expressed according to the Ising-like form $E\left(\eta \right)=-J\sum_{\left(i,j\right)}\eta_{i}\eta_{j}-H\sum_{i=1}^{N}\eta_{i}$ [33], where *J* represents the interaction energy between spins, and *H* is a parameter accounting for the dependence on the local spin $\eta_{i}$ (usually the magnetic field). Giving that sgn(ℓ)=−sgn(−ℓ), one has H=0 for all values of θ. From Equation (Equation 15), the ratio $w_{\ell i}\left(\eta^{i}\right)/w_{\ell i}\left(\eta \right)$ is then expressed according to the local detailed balance:(16)$$\frac{w_{\ell i}\left(\eta \right)}{w_{\ell i}\left(\eta^{i}\right)}=e^{-\beta_{\ell }\left[E\left(\eta^{i}\right)-E\left(\eta \right)\right]}.$$

We are now in a position to obtain the model’s thermodynamic properties. The time variation of the mean energy U=〈E(η)〉 is given by $dU/dt=\sum_{\ell }\Phi_{\ell }$, where $\Phi_{\ell }$ denotes the heat exchanged due to the *ℓ*-th thermal reservoir, given by
(17)$$\Phi_{\ell }=\sum_{i}\langle \left[E\left(\eta^{i}\right)-E\left(\eta \right)\right]w_{\ell i}\left(\eta \right)\rangle ,$$
constrained by the first law of thermodynamics, $\sum_{\ell }\Phi_{\ell }=0$ in the NESS. The entropy production and entropy flux are also decomposed into *ℓ*-indexed components by replacing $w_{i}\left(\eta \right)\to w_{\ell i}\left(\eta \right)$ in Equations (Equation 5) and (Equation 6). In particular, the latter reads
(18)$$\sigma_{\ell }=\sum_{\eta }p^{st}\left(\eta \right)\sum_{i}w_{\ell i}\left(\eta \right)\ln\frac{w_{\ell i}\left(\eta \right)}{w_{\ell i}\left(\eta^{i}\right)}.$$

Since the entropy change vanishes at the NESS, $dS/dt=\sum_{\ell }\left(\Pi_{\ell }-\sigma_{\ell }\right)$, both entropy production and entropy flux can be identified by Equation (Equation 18): $\Pi =\sum_{\ell }\Pi_{\ell }=\sum_{\ell }\sigma_{\ell }=\sigma$. The expressions above are consistent with Refs. [16,32].

Finally, by inserting Equation (Equation 16) into Equation (Equation 18), each entropy flux component $\sigma_{\ell }$ is related with exchanged heat $\Phi_{\ell }$ by a Clausius-like form $\sigma_{\ell }=-\beta_{\ell }\Phi_{\ell }$, where $\Phi_{\ell }$ is given by Equation (Equation 17). Alternatively, σ can also be written in the usual thermodynamics form as a sum of thermodynamic fluxes times forces:(19)$$\sigma =-\sum_{\ell }\beta_{\ell }\Phi_{\ell }\;\mathrm{or}\;\sigma =\sum_{\ell \neq 2}X_{\ell }\Phi_{\ell },$$
where the second temperature was set as a reference to define all (k/2)−1 thermodynamic forces $X_{\ell }\equiv \beta_{2}-\beta_{\ell }$, associated with its respective flux, $\Phi_{\ell }$. For simplicity, we set the Ising interaction parameter to J=1. From the expression for E(η), it follows that $\Delta E=2\eta_{i}\ell$, which can be rewritten as $\Delta E=2\eta_{i}\left|\ell \right|\mathrm{sgn}\left(\ell \right)$. By taking the logarithm of Equation (Equation 16), it follows that
(20)$$\ln\frac{w_{\ell i}\left(\eta \right)}{w_{\ell i}\left(\eta^{i}\right)}=-2\beta_{\ell }\left|\ell \right|\eta_{i}\mathrm{sgn}\left(\ell \right).$$

Since the transition rates associated with each thermal reservoir are mutually exclusive, a direct comparison with Equation (Equation 9) for a given *ℓ* provides to obtain each (reciprocal inverse) temperature $\beta_{\ell }$ given by
(21)$$\beta_{\ell }=\frac{1}{2|\ell |}H\left[\left|\ell \right|-\frac{k\theta }{1-\theta }\right]\ln\left(\frac{1-f}{f}\right),$$
where $\beta_{2}=2\beta_{4}=3\beta_{6}...=k\beta_{k}/2$ in the inertialess case. We pause to make a few comments. First, Equation (Equation 21) comes from the local detailed balance and constitutes one of the main results of this paper. Such temperature relation (see e.g., Equation (Equation 15)) provides a clear connection between entropy production and heat fluxes, in which (as shall be discussed later) some of the heat components has to be strictly positive and negative in order to ensure σ≥0. Second, Equation (Equation 21) extends the calculation of temperatures for a given neighborhood and inertia, and reduces to the expression from Ref. [19] as θ=0. Third, $\beta_{\ell }$ vanishes for large enough values of inertia $\theta >\theta_{p}$, illustrating that despite a heat flux associated with the *ℓ*-th reservoir being well-defined, it does not produce entropy. Fourth and last, the temperature assumes a different value for $\theta =\theta_{p}$ given by
(22)$$\beta_{\ell }=-\frac{1}{2|\ell |}\ln\left(2f\right).$$

This completes our description of the temperature definitions for the MV as well as the influence of inertia. Now we turn to unravel the role of each *ℓ* to the fluxes of heat and entropy production. Starting with the inertialess case, where $\beta_{2}>\beta_{4}>...>\beta_{k}$, we argue that the heat fluxes associated with the states in contact with the coldest and hottest baths are always positive and negative, respectively: $\Phi_{2}<0$ and $\Phi_{k}>0$, whose a (nonrigorous) argument is present as follows. Starting with the two thermal baths case (k=4), it is straightforward to verify that, since σ acquires the simple form $\sigma =\left(\beta_{2}-\beta_{4}\right)\Phi_{4}>0$. Given that $\beta_{2}-\beta_{4}>0$ (cf. Equation (Equation 21)), it follows that $\Phi_{4}\geq 0$ and hence $\Phi_{2}=-\Phi_{4}\leq 0$. The case of more than two reservoirs is more intriguing, since intermediate fluxes can be positive, negative, or even change their sign upon *f* being varied (see e.g., the first figure (d) in Section 3.3). For k=6, one has $\sigma =-\left(\beta_{2}-\beta_{6}\right)\Phi_{2}-\left(\beta_{4}-\beta_{6}\right)\Phi_{4}\geq 0$ and three possibilities for $\Phi_{2}$ and $\Phi_{4}$. The former, in which both are negative, promptly implies σ≥0, whereas the second case, $\Phi_{2}\leq 0$ and $\Phi_{4}\geq 0$, is also consistent since $-\left(\beta_{2}-\beta_{6}\right)\Phi_{2}\geq \left(\beta_{4}-\beta_{6}\right)\Phi_{4}$ and hence $\Phi_{6}\geq 0$ (recalling that $\Phi_{6}=-\left(\Phi_{2}+\Phi_{4}\right)$). The third possibility, in which $\Phi_{2}\geq 0$ and $\Phi_{4}\leq 0$ violates the second law in some cases and thus it is not possible. Similar findings are verified for θ≠0, but we should note that only neighborhoods with $\ell^{*}$ greater than ℓ−kθ/(1−θ) will contribute to the entropy production, $\sigma =-\sum_{\ell^{*}}^{k}\beta_{\ell }\Phi_{\ell }$. For example, for k=20 and distinct inertia intervals 3/8<θ≤7/17, 7/17<θ≤4/9, θ>4/9, the corresponding entropy productions read $\sigma =-\sum_{\ell =14}^{k}\beta_{\ell }\Phi_{\ell }$, $\sigma =-\beta_{16}\Phi_{16}-\beta_{18}\Phi_{18}-\beta_{20}\Phi_{20}$ and $\sigma =-\beta_{18}\Phi_{18}-\beta_{20}\Phi_{20}$, such latter one similar to the k=4 case (but here $\sum_{\ell =2}^{k}\Phi_{\ell }=0$) and once again illustrating that $\Phi_{\ell^{*}=18}\leq 0$ and $\Phi_{k=20}\geq 0$. We close this section by pointing out that, despite the above nonrigorous argument, the general finding $\Phi_{\ell^{*}}\leq 0$ and $\Phi_{k}\geq 0$ has been verified in all cases. In contrast, it is not possible to draw general conclusions about intermediate fluxes, in which some change sign as *f* increases.

### 3.2. Fluctuation Theorems

Thermodynamic consistent systems satisfy the detailed fluctuation theorem (DFT) for entropy production, which gives rise to the stochastic version of the second law. It states that negative fluctuations of the integrated entropy production are exponentially suppressed by the positive counterparts. For a given integration window τ, the DFT is asymptotically valid for $\Sigma =\int_{0}^{\tau }\sigma \left(t\right)dt$ at the NESS since it is equal to the entropy production:(23)$$\lim_{\tau \to \infty }\ln\frac{P_{\tau }\left(\Sigma \right)}{P_{\tau }\left(-\Sigma \right)}=\Sigma ,$$
where $P_{\tau }\left(\Sigma \right)$ represents the probability of measuring Σ in a trajectory of length τ. This relation holds beyond the long-time limit when the internal change of configuration entropy is considered in addition to the entropy fluxes. Consequence of the above, the integral fluctuation theorem (IFT) reads
(24)$$\lim_{\tau \to \infty }\langle {e^{-\Sigma }}_{\tau }\rangle =1$$
and is useful for relating the components of Σ, such as in the celebrated Jarzynski equality [34] that relates the statistics of work to free energy differences, bridging equilibrium and nonequilibrium quantities. The feasibility of employing such methods is tightly related to the ability to observe fluctuations in the trajectories, which become rare as τ increases. We explore the manifestation of these relations, cornerstones of stochastic thermodynamics, in the MV vote model.

The left panel of Figure 4 shows the convergence of the left-hand side of Equation (Equation 23) to its right-hand side as the integration windows become larger for the entropy production evaluated from Equations (Equation 18) and (Equation 19). Observing the DFT becomes an expensive task even for small systems since the negative fluctuations of entropy production become increasingly rare for larger values of τ. The right panel shows the left-hand side of the IFT in Equation (Equation 24), which converges to one despite the presence of inertia. It is worth mentioning that the convergence is observed from above and from below. Although no general conclusion can be drawn, the behavior of these fluctuation relations might be related to the phase transitions: In the examples, the IFT presents a slower convergence at the vicinity of the phase transition.

### 3.3. Heat Fluxes at Phase Transitions

According to Equation (Equation 17), every heat flux $\Phi_{\ell }$ is an ensemble average and, therefore, we expect at least the most significant components of the entropy production to behave similarly to σ at the vicinity of a phase transition. More specifically, at discontinuous phase transitions, the curves of entropy production cross at $f_{c}$ for distinct system sizes in regular lattices, and a hysteretic branch is present in complex topologies [13]. These properties are promptly verified for the largest fluxes ℓ=12, 14 and 20. For a regular lattice, panels (a)–(c) of Figure 5 display the crossing of the fluxes for different systems sizes, and panel (d) shows the quantitative value of each individual flux. For a random-regular network, panels (a)–(c) of Figure 6 show the hysteretic branch while (d) shows individual flux values.

The continuous lines in panels Figure 5a–c are obtained from the bimodal Gaussian description in Equation (Equation 11), in good agreement with the simulation results. Remarkably, for both regular and complex topologies, the phase transition can be probed and precisely located from the behavior of any individual flux.

### 3.4. Contributions to Dissipation

Inspecting the thermodynamic contribution of individual *ℓ*’s raises the question of how each type of neighborhood contributes to entropy production, a measure of dissipation. As previously discussed, the second law imposes $\Phi_{\ell^{*}}<0$ and $\Phi_{k}>0$ irrespective of *f*, and also local configurations satisfying $\left|\ell |<\right|\ell^{*}|$ do not dissipate. Taking into account that some intermediate fluxes $\Phi_{\ell }$ are nonmonotonic in terms of *f*, one could expect that they would present a less significant contribution. Inspired by evidence from simulations, we observe the predominance of $\Phi_{\ell^{*}}$ and $\Phi_{k}$, hence we introduce the contribution of these two fluxes as $\sigma_{\ell^{*},k}=-\beta_{\ell^{*}}\Phi_{\ell^{*}}-\beta_{k}\Phi_{k}>0$. This represents an approximation but not a bound since the remaining fluxes can change their signs.

Figure 7 compares, for the random-regular and regular lattices, $\sigma_{\ell^{*},k}$ and σ for distinct values of θ. In all cases, $\sigma_{\ell^{*},k}$ is not only close to σ but also captures the qualitative behavior, successfully describing the interplay between the control parameter *f*, inertia θ, and the dissipation, including a peak located at the vicinity of the phase transition. For larger θ the set of dissipating local configurations shrinks, hence the better agreement between both curves.

## 4. Conclusions

The nonequilibrium thermodynamic theory of the generic majority vote model was presented and thoroughly investigated, encompassing its phase transition. A consistent definition of temperature and the connection between heat fluxes and entropy production were introduced and analyzed in the context of continuous and discontinuous phase transitions. The present approach for fluxes is thermodynamically consistent and equivalent to the microscopic entropy production definition and satisfies the detailed fluctuation theorem.

We believe that the present framework not only conciliates the thermodynamic aspects of an important class of nonequilibrium systems but also introduces a new kind of nonequilibrium ingredient, based on the idea of a thermal reservoir associated with the system neighborhood. Such an idea has revealed general for a generic voter-like model with “up-down” $Z_{2}$ symmetry. In the presence of inertia, the spin changes induced by some local configurations are reversible. Moreover, we explore what are the most relevant neighborhoods driving the system dissipation, including its qualitative features across a phase transition, and how these neighborhoods contribute to the structure of the phase diagram.

Our findings are valid for a class that describes systems from social dynamics to the physics of thermal engines, presenting collective effects that can be leveraged for improved performance. Such potential application raises interesting questions such as the role of lattice topology and even the kind of voter model used (see e.g., Ref. [19] for a comparison between them) in order to optimize the desirable power and efficiency. Such topics should be investigated in the future. 

## Figures and Tables

**Figure 1 entropy-25-01230-f001:**
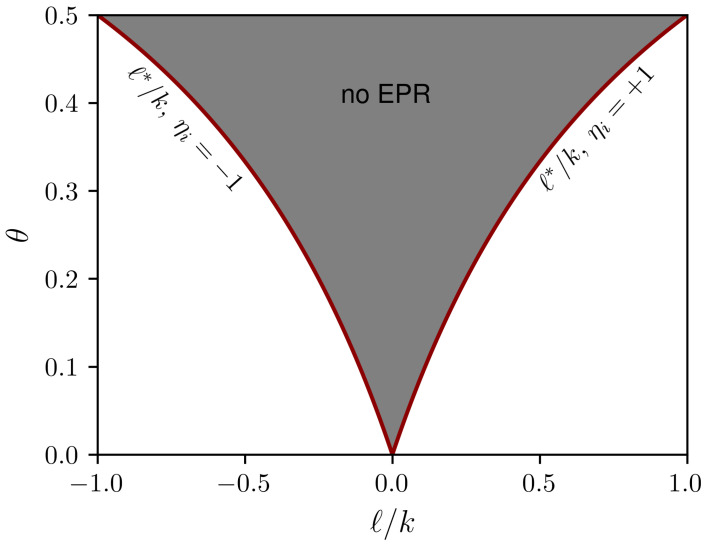
Scheme representing the values of $\ell^{*}$ for $\eta_{i}=\pm 1$ corresponding to the plateaus. In the shaded area, where $\left|\ell |<\right|\ell^{*}|$, the neighborhoods do not contribute to the entropy production.

**Figure 2 entropy-25-01230-f002:**
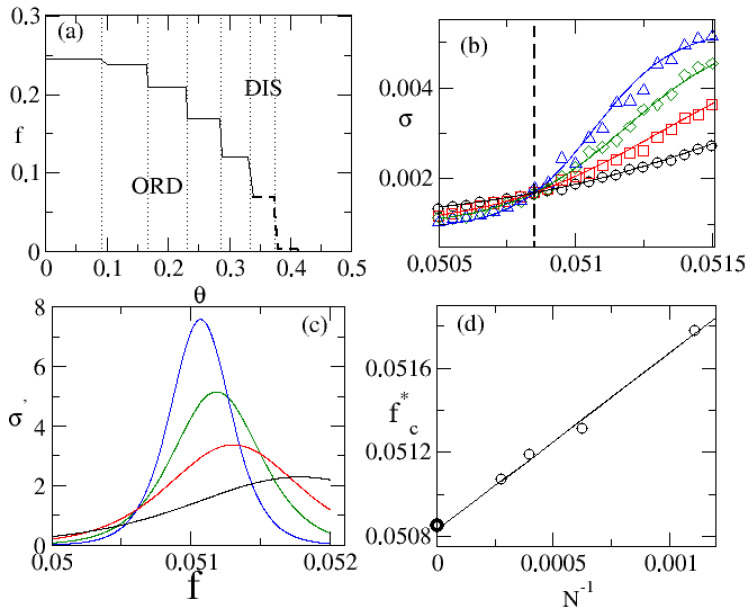
In (**a**), the phase diagram of the inertial majority model for a regular lattice for k=20. Vertical lines mark the plateau positions predicted in Equation (Equation 3). Panel (**b**) depicts the entropy production σ for distinct system sizes $N=L^{2}$’s. Continuous lines denote the phenomenological description from Equation (Equation 11) and vertical line corresponds to the crossing among entropy production curves at $f_{c}=0.05085\left(2\right)$. In (**c**), the derivative $\sigma^{\prime }\equiv d\sigma /df$ versus *f* obtained from continuous lines in (**b**). Panel (**d**) show the position $f_{c}^{*}$ of maximum of $\sigma^{\prime }$ versus $N^{-1}$ and its accordance with the crossing among entropy production curves yielding (symbol •) as N→∞.

**Figure 3 entropy-25-01230-f003:**
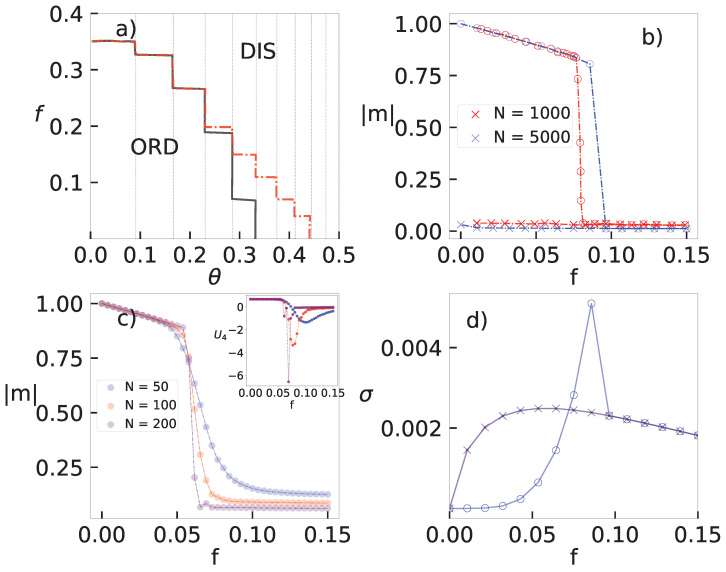
Phase transition for the MV in a random-regular topology with connectivity k=20. Panel (**a**) depicts the phase diagram θ versus *f*. Continuous and dashed lines show, for a system of size $N=10^{4}$. Note that a hysteretic branch for θ>3/13. Panels (**b**,**c**) show, for θ=3/8, the order parameter |m| versus *f* for distinct large and small system sizes *N*, respectively. Inset: the reduced cumulant $U_{4}$ versus *f*. Circles and × attempt to the forward and backward “trajectories”, respectively. In (**d**), its corresponding σ’s for N=5000.

**Figure 4 entropy-25-01230-f004:**
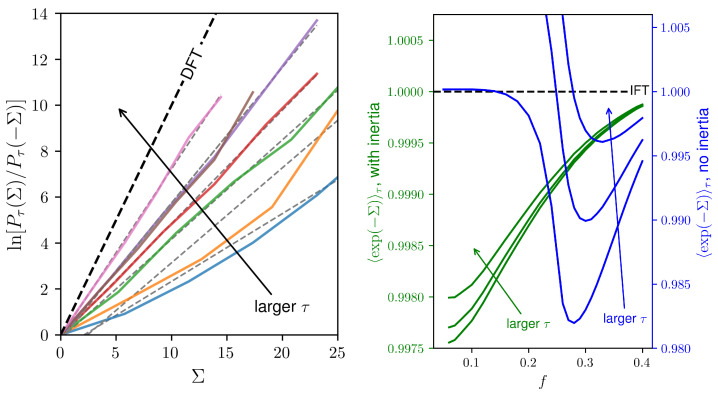
(**Left**) Convergence to the detailed fluctuation theorem as integration window τ increases for a lattice L=6 and f=0.04; solid lines are simulation results while dashed lines are the respective linear fits. (**Right**) Convergence to the integral fluctuation theorem for the case with no inertia (blue) and with inertia θ=3/8 (green); additional parameters are k=20 and $N=10^{4}$.

**Figure 5 entropy-25-01230-f005:**
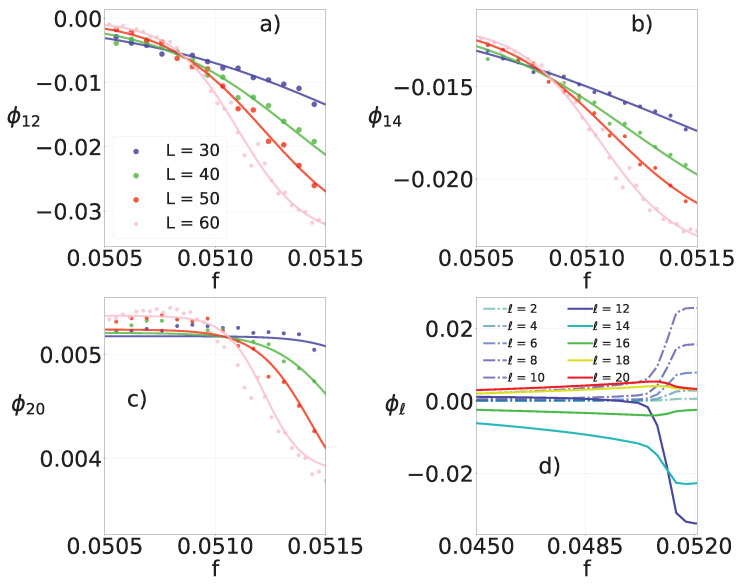
For the regular lattice with θ=3/8, k=20 and distinct system sizes $N=L^{2}$, panels (**a**,**b**) depict the most representative (largest absolute values) heat fluxes per particle $\Phi_{\ell }$’s versus control parameter *f*. Continuous lines denote correspond to the phenomenological approach according to the ideas of Equation (Equation 11). Although the component heat flux panel (**c**) mildly changes with *f*, all curves also cross at $f_{c}$. Panel (**d**) shows all $\Phi_{\ell }$’s (ℓ=2,4,...,k) for $N=60^{2}$.

**Figure 6 entropy-25-01230-f006:**
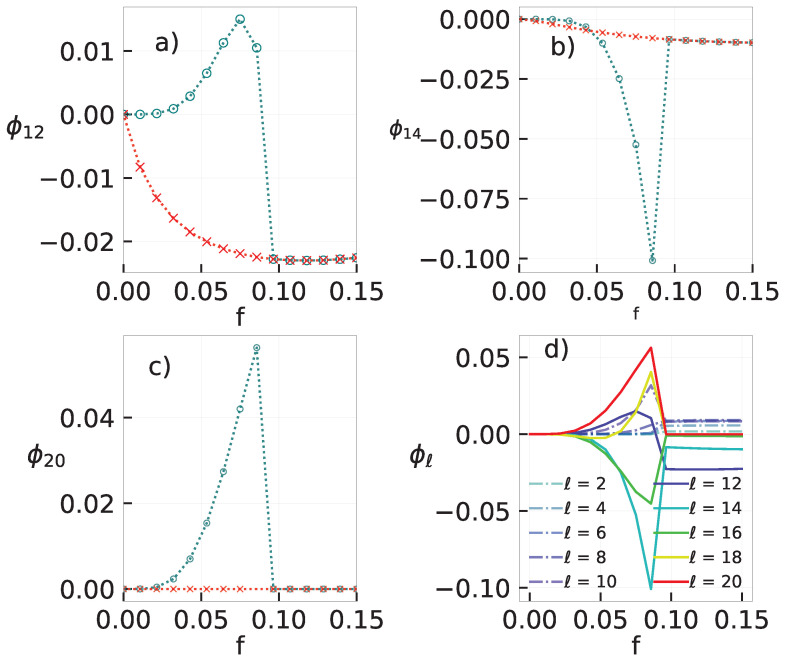
For a system of size N=5000, (**a**)–(**d**) the same as before, but for a random-regular structure.

**Figure 7 entropy-25-01230-f007:**
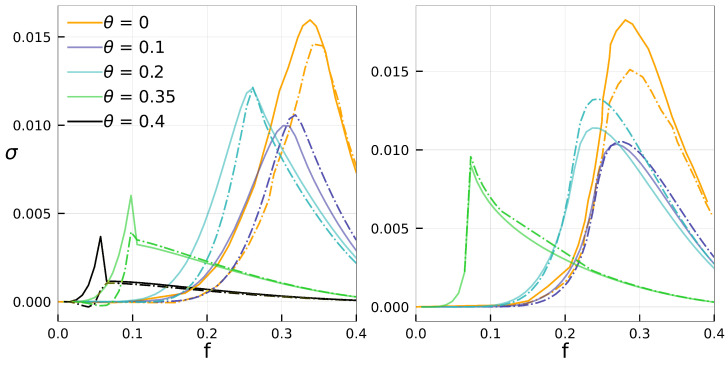
For k=20, random-regular (**left**) and regular (**right**) structures of sizes N=1600 and $40^{2}$, curves for $\sigma_{\ell^{*},k}$ (dot-dashed) and σ (continuous) are shown in terms of *f* for distinct θ’s. From top to bottom, $\ell^{*}=2,4,6,12$ and 14.

## Data Availability

Not applicable.

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
