# Peer review of "Nonequilibrium Thermodynamics of the Majority Vote Model"

_entropy, 2023, doi:10.3390/e25081230_

Round 1

Reviewer 1 Report

This article proposes non-equilibrium thermodynamics for a system that is not thermodynamic. It introduces a series of magnitudes analogous to thermodynamics, such as entropy production and temperature, but their meaning is not well-founded. In particular, the definition of temperature is artificial and it is not clear that it is helpful in establishing equilibrium conditions. The authors should better justify the proposed concepts so that the article can be reconsidered.

Author Response

This article proposes non-equilibrium thermodynamics for a system that is not thermodynamic. It introduces a series of magnitudes analogous to thermodynamics, such as entropy production and temperature, but their meaning is not well-founded. In particular, the definition of temperature is artificial and it is not clear that it is helpful in establishing equilibrium conditions. The authors should better justify the proposed concepts so that the article can be reconsidered.

ANSWER: Indeed, the original majority vote model (see e.g. Refs. [2-13], mainly Refs. [12,13]) is strictly defined in terms of probabilistic rules with no connection to thermodynamics, in such a way that no connection between heat and dissipation has been carried out previously. Whereas the field of nonequilibrium thermodynamics, booming since the 90's, has unmatched and insightful results that bring physical meaning to stochastic dynamics such as the unraveling of energy fluxes and fluctuation relations. Establishing a thermodynamic interpretation to dynamical systems yields a rich framework to describe its behavior and understand its limits. Our work aims at bridging the majority vote model, largely studied due to many of its properties, and the toolbox of stochastic thermodynamics.

In general, thermodynamic consistency is imprinted in transition rates through local detailed balance, thus we introduce a notion of effective thermal reservoirs mediating all types of jumps in the majority vote and, inspired by local detailed balance, establish effective temperatures. Despite their definition not being linked to actually existent reservoirs, our results provide another viewpoint for the majority vote model (and other voter-like models), as a nonequilibrium version of the Ising model. Furthermore, we study the fluxes of heat to the effective reservoirs, the behavior of nonnegative entropy production, and fluctuation relations.

Another remarkable point about the reliability of our approach to thermodynamics is that it provides a very clear connection between entropy production and heat fluxes, in which some of the heat components should be strictly positive and negative for ensuring the non-negativity of entropy production, which is the nonequilibrium version of the second law. Finally, we emphasize that our approach for the definition of temperatures is consistent with the entropy production satisfying well-known fluctuation relations, one of the main cornerstones of nonequilibrium thermodynamics.

In summary, we acknowledge the referee for pointing out that the previous version of the manuscript could be misleading with the physicality of reservoirs and the usefulness of connecting to thermodynamics. Therefore, we have improved the discussion including above arguments in the manuscript that can be found in red letters. In particular, see the updated Introduction and Section III.A.

Reviewer 2 Report

The authors have solved the thermodynamic problem in an ingenious way. The paper is elegant and has important cognitive value. Congratulations. I recommend that the paper be published in Entropy.

Author Response

We acknowledge the referee for a careful reading and pointing us out the novelty of our results. 

Reviewer 3 Report

The Authors propose a way of relating the theory of thermodynamic non-equilibrium to the generic majority vote model, defining a temperature and suggesting a way to relate heat fluxes and entropy. The work is interesting, well presented and honestly discussed, and this Reviewer believes the paper can be accepted for publication in Entropy in its present form. As a suggestion, 'thermal reservoir' ought to be used instead of 'thermal bath', or, worse still 'heat bath'.

Author Response

ANSWER: We acknowledge the referee for a careful reading. We have incorporated the referee's suggestion and now, in the revised version, the expression "thermal bath" was replaced for "thermal reservoir".